# Interventions to improve the quality of cataract services: protocol for a global scoping review

Miho Yoshizaki [iD] ,[1] Jacqueline Ramke [iD] ,[1,2] João M Furtado,[3] Helen Burn,[4] Stephen Gichuhi,[5] Iris Gordon,[1] Ada Aghaji,[6] Ana P Marques [iD] ,[1] William H Dean,[1,7] Nathan Congdon,[8,9] John Buchan,[1] Matthew J Burton[1,10]

For numbered affiliations see end of article.

**Correspondence to**
Miho Yoshizaki;
miho.yoshizaki@nhs.net

## ABSTRACT

**Introduction** Cataract is the leading cause of blindness globally and a major cause of vision impairment. Cataract surgery is an efficacious intervention that usually restores vision. Although it is one of the most commonly conducted surgical interventions worldwide, good quality services (from being detected with operable cataract to undergoing surgery and receiving postoperative care) are not universally accessible. Poor quality understandably reduces the willingness of people with operable cataract to undergo surgery. Therefore, it is critical to improve the quality of care to subsequently reduce vision loss from cataract. This scoping review aims to summarise the nature and extent of the published literature on interventions to improve the quality of services for primary age-related cataract globally.

**Methods and analysis** We will search MEDLINE, Embase and Global Health for peer-reviewed manuscripts published since 1990, with no language, geographic or study design restrictions. To define quality, we have used the elements adopted by the WHO—effectiveness, safety, people-centredness, timeliness, equity, integration and efficiency—to which we have added the element of planetary health. We will exclude studies focused on the technical aspects of the surgical procedure and studies that only involve children (<18 years). Two reviewers will screen all titles/abstracts independently, followed by a full-text review of potentially relevant articles. For included articles, data regarding publication characteristics, study details and quality-related outcomes will be extracted by two reviewers independently. Results will be synthesised narratively and presented visually using a spider chart.

**Ethics and dissemination** Ethical approval was not sought, as our review will only include published and publicly accessible information. We will publish our findings in an open-access peer-reviewed journal and develop an accessible summary of the results for website posting. A summary of the results will be included in the ongoing *Lancet Global Health* Commission on Global Eye Health.

**Registration details** Open Science Framework (https://osf.io/8gktz).

## Strengths and limitations of this study

► A strength of this review is the use of a broader concept of quality beyond the common measure of postoperative visual acuity—we included the seven elements of quality outlined in WHO's framework for healthcare quality, as well as the element of planetary health.

► Another strength is that we have broadened the scope of cataract services beyond the surgical intervention itself to identify interventions to improve quality along the care pathways, from detection and referral to uptake of services through to postoperative care.

► This study will not include studies that assess specific surgical techniques and/or specific products and medications as this extensive literature is commonly synthesised in Cochrane and other reviews.

► This review will summarise the nature and extent of the literature on interventions to improve the quality of cataract services but will not assess the quality or risk of bias of the studies themselves.

## INTRODUCTION

Cataract is the leading cause of blindness globally and a major cause of moderate and severe vision impairment—an estimated 65 million people had vision loss from cataract in 2015.[1] Vision loss from cataract is unequally distributed throughout the world. For example, in 2015, among adults aged 50 years and above, the age-standardised prevalence of cataract blindness ranged from 0.08% (80%, uncertainty interval (UI) 0.03%–0.19%) in high-income countries of the Asia Pacific region to 2.35% (80%, UI 0.72%–5.04%) in West sub-Saharan Africa—almost a 30-fold difference.[1] Inequality (ie, measurable differences between population subgroups) is also evident within countries, with a higher prevalence of cataract blindness among socially disadvantaged groups such as women, rural dwellers and those who are not literate.[2]

Cataract surgery is an efficacious intervention that can restore vision[3–5] and alleviate poverty.[6] It is one of the most common surgical interventions in many high-income

countries and some middle-income countries.[7] However, good quality services are not universally accessible, particularly in low/middle-income countries (LMICs).[8 9] Poor quality understandably reduces the willingness of people with operable cataract to undergo surgery.[10] Therefore, it is critical to improve the quality of care to subsequently reduce vision loss from cataract.

Quality of cataract services is most commonly measured using postoperative visual acuity. Measuring and monitoring outcomes is crucial in order to improve them[11] and tools are available to enable monitoring of postoperative visual acuity.[12]

Beyond using postoperative visual acuity to assess effectiveness, the quality of cataract services includes many clinical and non-clinical dimensions.[13] For example:

- ► Timeliness: cataract commonly occurs bilaterally. In many settings, the current recommendation is to operate on one eye at a time and allow enough time for the operated eye to heal before operating on the second eye. However, delay in surgery for the second eye has been linked to increased risk of falls and road traffic accidents.[14]
- ► People-centredness: it may be common for patients to have to visit hospitals several times before the surgery for different preoperative assessments, even though some of these could be done in one visit. Reducing the number of hospital visits to get surgery would improve quality from the patient perspective.
- ► Equity: there is no physiological reason why outcomes should be poorer in women compared with men, but women tend to have lower access and poorer postoperative vision outcomes compared with men.[2 15] A further example of inequity is seen in the difference in effective cataract surgical coverage among indigenous (51.6%, 95% CI: 42.4–60.7) and non-indigenous Australians (88.5%, 95% CI: 85.2–91.2).[16]
- ► Efficiency (productivity): there is a link between the quantity of surgery a surgeon performs and the quality of that surgery.[17] It has also been demonstrated that apparently cheaper service delivery options, such as outreach camps, can be less cost-effective compared with surgery delivered in static clinics due to worse outcomes.[18]

The aim of this review is to summarise the nature and extent of the published literature on interventions to improve the quality of cataract services globally. We chose to undertake a scoping review rather than an alternative evidence synthesis approach because we wished to identify and map the available evidence, which we anticipate will be heterogeneous.[19 20] We will take a broad perspective on quality outcomes and relevant interventions of interest, but will exclude studies focused exclusively on the technical aspects of surgical techniques. For example, we will not include studies reporting the effectiveness of phacoemulsification or manual small incision surgery, as these are summarised in other reviews.[3–5 21]

## Definitions and framework development

Cataract services include the range of activities on the pathway from detecting people with operable cataract to these people undergoing surgery and receiving postoperative care. As such, cataract services are both community and facility-based[22] and—regardless of the setting—should involve a broad range of healthcare providers from the community level (eg, village health workers as case finders) through primary (eg, optometrist) and secondary services (ie, surgical team). In addition, consideration of all of the health system building blocks is relevant to strengthen cataract services.

Quality of care is one of the objectives embodied by the concept of Universal Health Coverage, together with equity in access and financial protection.[23] Our review will be guided by the definition of the quality of care recently outlined by the WHO:

> Quality of care is 'the degree to which health services for individuals and populations increase the likelihood of desired health outcomes and are consistent with current professional knowledge'.[24]

WHO has adopted the framework of quality outlined by the Institute of Medicine.[25] This framework measures the quality of healthcare across seven elements, namely, effectiveness, safety, people-centredness, timeliness, equity, integration and efficiency.

We have made one addition to the quality elements in WHO's framework—we believe that *planetary health* is an essential element of quality cataract surgery, so will also scope the literature on this. Planetary health is focused on sustainability, including the ability of the society to make choices while balancing the needs of future generations.[26] This modified framework is shown in figure 1.

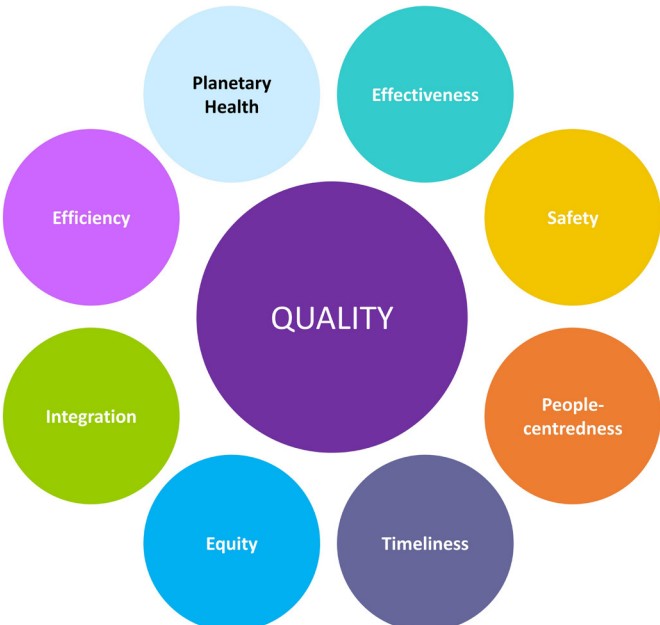

**Figure 1** Elements of healthcare quality considered in this review (modified figure 3.2 from World Health Organization[24] by adding *planetary health*).

To help guide the scope of our review, we mapped examples of outcome measures and interventions for cataract services against each of the eight elements of healthcare quality (table 1). These outcomes and interventions were drawn from the literature,[27][28] as well as the knowledge and experience of the authorship group. For people-centredness, we drew on the outline of Integrated Person-Centred Health Services provided by WHO and adopted in the recent *World Report on Vision,* whereby services aim to provide coordinated care that addresses the full spectrum of eye conditions according to an individual's needs and recognises people as participants and beneficiaries of this care.[29][30]

When mapping interventions, we categorised them using the WHO health systems 'building blocks', that is, we mapped them to the most relevant of service delivery; health workforce/human resources (HR), health information system (HIS); access to essential consumables/non-consumables; financing; and leadership/governance. Recognising that this framework does not include community engagement and empowerment, we added *community* as an additional category against which interventions could be mapped.[31]

## METHODS AND ANALYSIS
### Objectives/scoping review questions
We aim to answer the following three questions:
1. What interventions to improve the quality of cataract services have been described in the published literature?
2. Which element(s) of quality did the interventions address?
3. Where was the evidence generated (high-income vs middle-income vs low-income settings)?

### Protocol and registration
This protocol for this scoping review is reported according to the relevant sections of the Preferred Reporting Items for Systematic Reviews and Meta-Analyses Extension for Scoping Reviews (PRISMA-ScR) guideline (online supplementary annex 1).[32]

### Eligibility criteria
This scoping review will include primary research studies of any design and systematic reviews from any country that report a quality-relevant outcome for primary age-related cataract following an intervention related to the quality of cataract services. We will only include studies where intervention is compared against any alternatives (eg, intervention vs no intervention/current practice vs new intervention/before vs after implementation). Examples of relevant interventions are provided in table 1, mapped against the eight quality elements of interest. Systematic reviews will be included only if meta-analysis is conducted for a quality-relevant outcome. If we identify systematic reviews that report narrative synthesis of quality-relevant outcomes without meta-analysis, then we will review the

list of included studies and include in our scoping review any that meets our eligibility criteria.

We will exclude studies assessing specific surgical techniques (eg, phacoemulsification vs manual small incision surgery, site of anaesthesia and size of incision) and/or specific products and medications used during and around the time of surgery (eg, monofocal vs multifocal intraocular lens; drug A vs drug B) as these are typically addressed in other systematic reviews.[3][4][21] Studies focused exclusively on cataract services for children (aged under 18 years) will be excluded, as these services differ substantially from those for age-related cataract. We will also exclude studies reporting interventions to prevent cataract formation or progression. We will exclude studies published prior to 1990, as during the last 30 years, there have been a large number of major developments in cataract services that would be expected to have changed the 'landscape' substantially. Service delivery models prior to this time are quite different from those currently used. There will be no language limitations. Only studies where the full text is available will be included.

### Search
We will search MEDLINE, Embase and Global Health databases using search strategies developed by a Cochrane Eyes and Vision Information Specialist (IG). The search strategy for MEDLINE is included in the online supplementary annex 2. We will examine reference lists of all included articles to identify further potentially relevant reports of studies. Field experts will be provided a list of the included studies and requested to identify further potentially relevant studies for consideration in the review.

### Selection of sources of evidence
Covidence systematic review software will be used for screening (Veritas Health Innovation, Melbourne, Australia. Available at: www.covidence.org). Each title and abstract will be screened independently by two reviewers (MY, JR, HB, AA, JB, JF, SG and WHD) to exclude publications that clearly do not meet the inclusion criteria. Subsequently, the full-text article will be retrieved for review if the citation seems potentially relevant and two reviewers will independently assess each article against the inclusion and exclusion criteria. Any discrepancies between the reviewers will be resolved by discussion and a third reviewer will be consulted if necessary. A PRISMA flow diagram will be completed to summarise the study selection process.

### Data charting process
A custom form will be developed in Excel for data charting. The form will be piloted on three studies and required amendments agreed by consensus. We anticipate a broad scope of included studies, so data charting will be an iterative process throughout the review and the data charting form will be amended as required. Each included study will be charted independently by two

**Table 1** Indicative outcomes and interventions to improve the quality of cataract services (excluding technical aspects of surgery and anaesthesia, equipment and medication).

| Quality elements | Description/notes | Example outcome measures | Example interventions |
|---|---|---|---|
| Effectiveness | WHO's framework defines this as adherence to evidence-based medicine.[24] | ▲ Effective cataract surgical coverage[15]<br>▲ Pre and post-operative visual acuity.<br>▲ Contrast, glare, colour vision.<br>▲ Years of sight-loss avoided. | ▲ Service delivery: day case vs inpatient surgery; risk stratification of patients and matching with surgeon skills.<br>▲ Equipment/consumables: preoperative biometry correctly undertaken and interpreted; access to a good quality range of intraocular lens (IOL) powers.<br>▲ HIS: recording and monitoring of outcomes—national data reporting system eg, cataract surgery minimum dataset in the UK and annual audit based on these data,[36] PRECOG;[37] BOOST;[12] national benchmarks for quality outcomes; post-operative spectacle supply. |
| Safety | Patient harm is the 14th leading cause of global disease burden.[24] | ▲ Wrong lens insertion<br>▲ Postoperative issues, for example, endophthalmitis, cystoid macular oedema, retinal detachment, corneal oedema and decompensation incidents<br>▲ Refractive outcomes, for example, target spherical equivalent, prediction error and postoperative astigmatism. | ▲ Service delivery: interventions to address surgical complications; protocols for emergency management of postoperative complications; post-operative care.<br>▲ HR: simulation training; continuing professional development for ophthalmologists.<br>▲ HIS: system to monitor individual surgeon performance.<br>▲ Governance: national benchmarks for quality outcomes in place (including refraction) quality assurance practice (ie, WHO cataract checklist and monitoring of outcomes).<br>▲ Equipment/consumables: IOL quality control, instrument sterilisation. |
| People-centredness | A good quality service should systematically incorporate the needs and preferences of patients. | ▲ Patient Reported Outcome Measures for example, EQ-5D, Catquest-9SF, Visual Function Questionnaire (VFQ-25).<br>▲ Number of hospital attendances required. | ▲ Community: counselling about accessing surgery; informed consent process; social support (eg, escort, family permission/support); dedicated eye health coordinators; preoperative anxiety reduction strategies. |
| Timeliness | Timely access to cataract surgery would improve patients' experience and reduce the risk of complications. Early identification and appropriate referral is key to timely access. | ▲ Severity of cataract at first presentation (including bilateral or unilateral).<br>▲ Time from diagnosis with operable cataract to completion of surgery.<br>▲ Inter-operative time for patients with bilateral cataract. | ▲ Service delivery: re-design of pathways (diagnostics, referrals, treatment and follow-up) to be acceptable, affordable and sustainable; use of technology for example, telemedicine; same-day bilateral surgery in low population density, low infection setting; strategies to reduce the waiting list. |
| Equity | Quality of care should not vary within the same setting according to patients' characteristics such as age, gender, ethnicity, rural/urban and socioeconomic status. Equity can be considered in terms of equity of access to healthcare services or equity of health outcomes. | ▲ Prevalence of cataract blindness and vision impairment in subpopulation (eg, gender, ethnic minority and indigeneity).<br>▲ Volume, distribution and effective coverage of surgery in subpopulations. | ▲ Service delivery: outreach diagnostic protocols including consideration for false positives/negatives.<br>▲ Equipment/consumables: reduced tax on imported items.<br>▲ Community: financial support for patients who need it (ie, subsidy for surgery and transport); patient information and education to raise awareness/anxiety management.<br>▲ Financing: health insurance for cataract surgery.[38] |

Continued

**Table 1** Continued

| Quality elements | Description/notes | Example outcome measures | Example interventions |
|---|---|---|---|
| Integration | Continuity of care and care coordination, including coordinating care for effectively managing comorbidities Improve the care experience for people. | ▲ Referral pathways.<br>▲ Multidisciplinary team training, accreditation and governance structure. | ▲ Service delivery: pathways (diagnostics, treatment and follow-up); support service; outreach and primary care screening diagnostic protocols / algorithms including consideration for false positives/negatives. |
| Efficiency | Efficient use of resources, including productivity of surgeons, would contribute to quality improvement at population level. Health service efficiency can be considered as allocative efficiency (optimal mix of inputs is being used to produce chosen outputs that is, multi-disciplinary team, financial allocation) and technical efficiency (ie, productivity of surgeons etc). | ▲ Productivity of surgeons (ie, annual cataract operations per surgeon).<br>▲ Availability of manager/administrator.<br>▲ Multidisciplinary fixed/permanent team.<br>▲ Financial management.<br>▲ Cost-effectiveness analysis. | ▲ HR: multidisciplinary team to support the surgeon, for example, nurses seeing post-operative patients; task-shifting to non-ophthalmologist cataract surgeons; eye department manager; removing the need for a specialist anaesthetist.<br>▲ Financing: financial sustainability of the providers; eye department autonomy over funds (budget and/or bank account); payment options that incentivise productivity and quality improvement (ie, fee per service and bundled payment); modelling of cost recovery options that balance productivity, affordability and profit.<br>▲ Equipment/consumables: dedicated operating theatre. |
| Planetary health | Healthcare is a major consumer of energy and resources and produces considerable amounts of emissions and waste. In order to protect and improve the health and well-being of future generations, it needs to shift towards an environmentally sustainable system. | ▲ Carbon footprint of cataract surgery.<br>▲ Waste generated during cataract surgery. | ▲ Equipment/consumables: reusable equipment, waste management.<br>▲ HIS: audit, lifecycle assessment.<br>▲ Financing: sustainable procurement. |

HIS, health information system; HR, human resources.

reviewers. Any discrepancies will be resolved by discussion and a third reviewer will be consulted if necessary. We plan to contact study authors in the case of unclear information and will make up to three attempts by email.

## Data items

The following data items will be collected during the data charting process:

1. Publication characteristics: title, year of publication, study design, country of origin and study setting.
2. Characteristics of intervention/study:
   a. Context (eg, geographic area, target population and distribution, type of interventions (categorised by health system building block), target health practitioner and duration/frequency).
   b. Quality element(s) addressed by the intervention (as outlined in table 1).
3. Outcome(s) of the intervention/study and whether it was reported to be effective (ie, had an effect vs had no effect) (examples of outcomes are outlined in table 1).

## Synthesis of results

We recognise that the indication for surgery can vary across different settings due to the prevalence of vision loss from cataract, the capacity of services and the quality and safety standards in each setting. Accordingly, we will synthesise results by World Bank country income level (high/upper-middle/lower-middle/low)[33] and (if possible) by Global Burden of Disease super-region (high income/Latin America and Caribbean/sub-Saharan Africa/North Africa and Middle East/Southeast Asia, East Asia and Oceania/South Asia/Central Europe, Eastern Europe and Central Asia).[34]

We will summarise findings narratively and using descriptive statistical methods as appropriate. We will map each intervention to the relevant quality element. We will visualise the findings using spider charts to show the extent of the evidence across each quality element and will plot evidence in high-income countries separately to LMICs. For each intervention, we will quantify the number of studies that were reported by the authors to be effective (vs having no effect).

## Patient and public involvement statement

This protocol was developed with input from the Commissioners of the *Lancet Global Health* Commission on Global Eye Health,[35] which includes people with lived experience of vision impairment (and cataract surgery), policymakers, academics, clinicians, government eye health programme leaders and advocacy specialists.

## ETHICS AND DISSEMINATION

Ethical approval was not sought, as our review will only include published and publicly accessible information.

We will publish our findings in an open-access, peer-reviewed journal and develop an accessible summary of the results for website posting and stakeholder meetings.

A summary of the results will also be included in the ongoing *Lancet Global Health* Commission on Global Eye Health.[35]

**Author affiliations**
[1]International Centre for Eye Health, London School of Hygiene and Tropical Medicine, London, UK
[2]School of Optometry and Vision Science, The University of Auckland, Auckland, New Zealand
[3]Division of Ophthalmology, Universidade de São Paulo, Faculdade de Medicina de Ribeirão Preto, Ribeirao Preto, São Paulo, Brazil
[4]Department of Ophthalmology, Stoke Mandeville Hospital, Aylesbury, UK
[5]Department of Ophthalmology, University of Nairobi, Nairobi, Kenya
[6]Department of Ophthalmology, University of Nigeria, Nsukka, Enugu, Nigeria
[7]Department of Ophthalmology, University of Cape Town, Rondebosch, Western Cape, South Africa
[8]Centre for Public Health, Queen's University Belfast, Belfast, UK
[9]Zhongshan Ophthalmic Center, Sun Yat-Sen University, Guangzhou, Guangdong, China
[10]Moorfields Eye Hospital, London, UK

**Contributors** JR and MJB conceived the idea for the review. MY and JR drafted and revised the protocol with suggestions from MJB, NC, JF, SG, HB, AA, APM, WHD and JB. IG constructed the search.

**Funding** MJB is supported by the Wellcome Trust (207472/Z/17/Z). JR is a Commonwealth Rutherford Fellow, funded by the UK government through the Commonwealth Scholarship Commission in the UK. The Lancet Global Health Commission on Global Eye Health is supported by The Queen Elizabeth Diamond Jubilee Trust, Moorfields Eye Charity [grant number GR001061], NIHR Moorfields Biomedical Research Centre, Wellcome Trust, Sightsavers, The Fred Hollows Foundation, The SEVA Foundation, British Council for the Prevention of Blindness and Christian Blind Mission.

**Competing interests** None declared.

**Patient and public involvement** Patients and/or the public were involved in the design, or conduct, or reporting, or dissemination plans of this research. Refer to the Methods section for further details.

**Patient consent for publication** Not required.

**Provenance and peer review** Not commissioned; externally peer reviewed.

**ORCID iDs**
Miho Yoshizaki http://orcid.org/0000-0003-1893-5675
Jacqueline Ramke http://orcid.org/0000-0002-5764-1306
Ana P Marques http://orcid.org/0000-0001-8242-7021

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
