## [Reviewer comments · BMJ Open]

ARTICLE DETAILS

TITLE (PROVISIONAL)	Interventions to improve quality of cataract services: protocol for a global scoping review
AUTHORS	Yoshizaki, Miho; Ramke, Jacqueline; Furtado, João; Burn, Helen; Gichuhi, Stephen; Gordon, Iris; Aghaji, Ada; Marques, Ana Patricia; Dean, William; Congdon, Nathan; Buchan, John; Burton, Matthew J

VERSION 1 - REVIEW

REVIEWER	Jennifer Hilgart Scientific Resource Center Portland VA Research Foundation USA
REVIEW RETURNED	10-Feb-2020

GENERAL COMMENTS	This scoping review protocol has been registered, has a stated aim and review questions. I suggest the authors provide a little more detail about their inclusion/exclusion criteria relating to study design. For example it states that 'primary research studies and systematic reviews' will be included. Will this include any study design? Also how will the systematic reviews be assessed (i.e. will they need to be of sound methodological quality? are narrative reviews included?) and how will systematic reviews be used (e.g. by extracting data about the number of relevant included studies?) Some description of the bubble diagram in Figure 2 would be helpful to show how the authors intend to present their findings. For example, what do the size of the bubbles represent? The number of studies within each domain? Are studies grouped by intervention or outcome? Page 12, line 20-21. I would suggest against conducting a meta-analysis for a scoping review - see the PRISMA-ScR explanation document - when no quality or risk of bias assessment is being conducted. Scoping reviews do not generally synthesize findings and it is not stated that you will extract data on the study findings in the data charting section. Also your review objectives do not relate to reporting on the effectiveness of interventions but rather to summarize the nature and extent of the literature. I found review question 4 a little confusing - do the authors intend to include an assessment of the effectiveness of the interventions or report on the volume of evidence? If the authors do intend to conduct a meta-analysis, more information is needed - see PRISMA-P checklist
---

	Spell out acronyms the first instance they are mentioned. For example, "IOL" in Table 1. Could there be further references to support the chosen method of scoping review - and/or any guidance used such as that from the Joanna Briggs Institute
--	--

REVIEWER	Assit.Prof. Thus Sanguansak KKU eye centre, Srinagarind hospital, Faculty of medicine, Khon kaen university, THAILAND Refractive cataract surgery and Retina Public health
REVIEW RETURNED	13-Feb-2020

GENERAL COMMENTS	Good review the data to know the problem in each elements of cataract surgery.
--

REVIEWER	Deanna Taylor City, University of London, England
REVIEW RETURNED	19-Feb-2020

GENERAL COMMENTS	Thanks you for inviting me to review this scoping review protocol on interventions to improve quality of cataract services. This is a well written manuscript and its subject matter is pertinent. The authors highlight that there is more to cataract service quality than simply post-operative visual acuity – this, I believe, is a particularly important message. My comments on this article are outlined below.  • The authors state that the review will exclude studies pertaining to the intra-operative period. Rather, they will limit included studies to those relating to the non-operative period. Yet, in Table 1, 'Intraoperative issues' are listed as a relevant outcome measure. • Would it be possible to provide any example outcome measures for 'Planetary Health' in Table 1? • The justification given for not involving patients and public is that this is a scoping review. However, patient and public involvement is certainly possible in a scoping review. I would therefore recommend that lack of patient and public involvement is discussed a limitation to this study.
---

VERSION 1 – AUTHOR RESPONSE

Reviewer 1:

I suggest the authors provide a little more detail about their inclusion/exclusion criteria relating to study design. For example it states that 'primary research studies and systematic reviews' will be included. Will this include any study design? Also how will the systematic reviews be assessed (i.e. will they need to be of sound methodological quality? are narrative reviews included?) and how will systematic reviews be used (e.g. by extracting data about the number of relevant included studies?)

Response: Thank you very much for your helpful comments and suggestions – we have provided additional clarification to the “Eligibility criteria” in response to your comments. We will not be

excluding systematic reviews based on their methodological quality as our aim is to map out existing evidence. The text on the eligibility criteria in the methods on page 9 has been updated as follows

“We will only include studies where an intervention is compared against an alternative (e.g. intervention vs. no intervention / current practice vs. new intervention / before vs. after implementation). Examples of relevant interventions are provided in Table 1, mapped against the eight quality elements of interest. Systematic reviews will be included only if meta-analysis is conducted for a quality-relevant outcome.”

Some description of the bubble diagram in Figure 2 would be helpful to show how the authors intend to present their findings. For example, what do the size of the bubbles represent? The number of studies within each domain? Are studies grouped by intervention or outcome?

Response: Thank you for this useful suggestion. We have added some additional information to our synthesis plan to clarify our approach. After careful consideration, we think that the bubble chart may not be the best way to present our analysis and we will rather use a spider chart. We have made this change in the abstract and synthesis of results sections.

Page 12, line 20-21. I would suggest against conducting a meta-analysis for a scoping review - see the PRISMA-ScR explanation document - when no quality or risk of bias assessment is being conducted. Scoping reviews do not generally synthesize findings and it is not stated that you will extract data on the study findings in the data charting section. Also your review objectives do not relate to reporting on the effectiveness of interventions but rather to summarize the nature and extent of the literature. I found review question 4 a little confusing - do the authors intend to include an assessment of the effectiveness of the interventions or report on the volume of evidence? If the authors do intend to conduct a meta-analysis, more information is needed - see PRISMA-P checklist

Response: This is a very helpful point for clarification. We did not plan to conduct a meta-analysis as part of this process, but rather wanted to flag our intention to undertake a future analysis should we find sufficient evidence (and for which we would generate an additional protocol). We agree this could potentially create confusion, so have deleted that sentence from the ‘Synthesis of results’ section.

Regarding data items on study findings, we have added details to item 3 i.e.

“Outcome(s) of the intervention/study and whether it was reported to be effective (i.e. had an effect versus had no effect) (examples of outcomes outlined in Table 1).”

Regarding the 4th review question, we appreciate this feedback. We have deleted the question and explained our intention to summarise the reporting of this information (rather than giving an overall estimate of effectiveness) in the ‘Synthesis of results’ section.

“For each intervention, we will quantify the number of studies that were reported by the authors to be effective (versus having no effect).”

Spell out acronyms the first instance they are mentioned. For example, "IOL" in Table 1.

Response: This has been addressed and updated in Table 1 of the revised manuscript.

Could there be further references to support the chosen method of scoping review - and/or any guidance used such as that from the Joanna Briggs Institute

Response: Thank you for this suggestion. We reviewed the JBI reviewer's manual which reinforced our chosen method of the scoping review. We have added this as further reference.

Reviewer: 2

Good review the data to know the problem in each elements of cataract surgery.

Response: Thank you for this positive feedback.

Reviewer: 3

- The authors state that the review will exclude studies pertaining to the intra-operative period. Rather, they will limit included studies to those relating to the non-operative period. Yet, in Table 1, 'Intraoperative issues' are listed as a relevant outcome measure.

Response: Thank you very much for your helpful feedback and suggestions. We will exclude studies where the intervention involves a specific surgical technique and/or a specific product or medication during and around the time of surgery because these are typically addressed in other systematic reviews. We identified 'intraoperative issues' as potential outcome measures because some interventions that are not specific to surgical techniques and/or specific products (e.g. introducing a surgical checklist) may address quality outcomes relating to intraoperative issues (e.g. wrong lens insertion). We appreciate this may introduce confusion, so have removed 'intraoperative issues' from the outcomes column.

- Would it be possible to provide any example outcome measures for 'Planetary Health' in Table 1?

Response: We added two examples to Table 1 of potential outcome measures for 'Planetary Health' (Carbon footprint of cataract surgery, Waste generated during cataract surgery).

- The justification given for not involving patients and public is that this is a scoping review. However, patient and public involvement is certainly possible in a scoping review. I would therefore recommend that lack of patient and public involvement is discussed a limitation to this study.

Response: On reflection, we realised we engaged a broad range of stakeholders, so have modified the text as below:

"This protocol was developed with input from the Commissioners of the Lancet Global Health Commission on Global Eye Health, which includes people with lived experience of vision impairment (and cataract surgery), policy makers, academics, clinicians, government eye health programme leaders and advocacy specialists."

VERSION 2 – REVIEW

REVIEWER	Jennifer Hilgart Scientific Resource Center Portland VA Research Foundation USA
REVIEW RETURNED	06-Apr-2020

GENERAL COMMENTS	Thank you for your revised manuscript. I think the word 'outcome' is missing from pg.9 of the manuscript in the 'Eligibility criteria' section. Its now stated that systematic reviews without a meta-analysis will be excluded. You may find systematic reviews where a meta-analysis of relevant outcome data is not possible and a narrative synthesis is reported. These systematic reviews would still be relevant to your scoping review?
---

VERSION 2 – AUTHOR RESPONSE

Reviewer 1:

Thank you for your revised manuscript.

I think the word 'outcome' is missing from pg.9 of the manuscript in the 'Eligibility criteria' section.

Response: Thank you for spotting this. We have amended the manuscript.

It's now stated that systematic reviews without a meta-analysis will be excluded. You may find systematic reviews where a meta-analysis of relevant outcome data is not possible and a narrative synthesis is reported. These systematic reviews would still be relevant to your scoping review?

Response: Thank you, this is a very helpful point for clarification. We have added some additional information to our eligibility criteria as below:

“If we identify systematic reviews which report narrative synthesis of quality-relevant outcomes without meta-analysis, we will review the list of included studies and include in our scoping review any that meet our eligibility criteria